# Disentangled Representation Learning with Sequential Residual Variational Autoencoder

## Abstract

Recent advancements in unsupervised disentangled representation learning focus on extending the variational autoencoder (VAE) with an augmented objective function to balance the trade-off between disentanglement and reconstruction. We propose *Sequential Residual Variational Autoencoder* (SR-VAE) that defines a "Residual learning" mechanism as the training regime instead of the augmented objective function. Our proposed solution deploys two important ideas in a single framework: (1) learning from the residual between the input data and the accumulated reconstruction of sequentially added latent variables; (2) decomposing the reconstruction into decoder output and a residual term. This formulation encourages the disentanglement in the latent space by inducing explicit dependency structure, and reduces the bottleneck of VAE by adding the residual term to facilitate reconstruction. More importantly, SR-VAE eliminates the hyperparameter tuning, a crucial step for the prior state-of-the-art performance using the objective function augmentation approach. We demonstrate both qualitatively and quantitatively that SR-VAE improves the state-of-the-art unsupervised disentangled representation learning on a variety of complex datasets. [1]

## 1 Introduction

Learning a sparse and interpretable representation of data is a critical component of a generalized, robust and explanatory intelligent system. This concept is inspired by human's ability to generalize the knowledge with abstract concepts and use them to reason the unseen environments Gupta et al. (2018). Despite recent advances on representation learning, it was shown that deep *convolutional neural networks* (CNN's) have a tendency to learn superficial statistics of data associated with given tasks, rather than important generative factors embedded in the physical world Jo & Bengio (2017); Goodfellow et al. (2014). One way towards this goal is disentangled representation learning which aims to capture the independent and interpretable generative factors of the data. Bengio et al. (2013) defines the disentangled representation intuitively as a representation where changes in one dimension correspond to changes in only one generative factor of the data, while being relatively invariant to changes in other factors. Recently, Higgins et al. (2018) assigned a principled definition by connecting symmetry transformations to vector representations using the group and representation theory.

Based on these definitions, disentangled representation can be learned in a supervised fashion where explicit and/or implicit prior knowledge on the generative factors of data are available. However, it is ideal to achieve this in an unsupervised learning setting to take advantage of the large amount of available unlabeled data. Along with the recent development of the generative models, many unsupervised disentangled learning approaches have been proposed based on either the generative adversarial networks (GAN) (proposed as InfoGAN in Chen et al. (2016)) or the variational autoencoders (VAE) (proposed as $\beta$-VAE in Higgins et al. (2017)). While $\beta$-VAE achieves better results and does not suffer from the training stability issue of InfoGAN, it faces a trade-off between the disentanglement and reconstruction due to its information bottleneck. The current state-of-the-art approaches extend the $\beta$-VAE with augmented objective function to reduce this trade-off Burgess et al. (2017); Kim & Mnih (2018); Chen et al. (2018); Kumar et al. (2017). A recent study by Locatello

---

[1]Codes available at: https://www.dropbox.com/s/5hkfn8xy5r8w5sz/Code.zip?dl=0

et al. (2018) carefully compared these approaches based on extensive experiments. They found that the performance of these approaches is very sensitive to the hyperparameter tuning associated with the augmented objective function and the initial random seed during training. More importantly, they proved that unsupervised learning of disentangled representation is impossible without introducing inductive bias on either the model or the data. We believe the trade-off between disentanglement and reconstruction in VAE-based approaches can be addressed by a different training approach. The idea of relying on modified training approaches, instead of augmented objective function, to encourage network behavior is commonly used for different problems. Take model over-fitting prevention for example, one way to address this is to augment the objective function with regularization terms, such as $L^1$ or $L^2$ regularization. An alternative solution is to apply special operations during training to enforce the generalization of the network representations, such as Dropout Srivastava et al. (2014) or Batch Normalization Ioffe & Szegedy (2015).

Our main contribution in this work is four-fold: 1) We propose *Sequential Residual Variational Autoencoder* (SR-VAE) that uses a novel "Residual learning" mechanism to learn disentangled representation with the original VAE objective. This is different from previous VAE-based approaches that merely focus on objective function design where hyperparameter tuning is crucial. 2) We show the proposed "Residual learning" mechanism defines an explicit dependency structure among the latent variables via sequential latent variable update. This encourages learning the disentangled representation. 3) We highlight that SR-VAE decomposes the reconstruction into residual and network decoder output via skip connection. This relaxation of reconstruction reduces the trade-off between disentanglement and reconstruction of VAE. 4) We demonstrate both qualitatively and quantitatively that SR-VAE improves the current state-of-the-art disentanglement representation learning performance on a variety of complex datasets.

## 2    CHALLENGES OF DISENTANGLING WITH AUGMENTED VAE OBJECTIVE

In this section, we first briefly review the VAE framework, followed by the $\beta$-VAE and its extensions for disentangled representation learning. We highlight the challenges of using an augmented objective function to balance the VAE's trade-off between the disentanglement and reconstruction. From these discussions, we then motivate the proposed SR-VAE framework.

VAE is a deep directed graphical model consisting of an encoder and a decoder Kingma & Welling (2013). The encoder maps the input data $x$ to a latent representation $q_\theta(z|x)$ and the decoder maps the latent representation back to the data space $q_\phi(x|z)$, where $\theta$ and $\phi$ represent model parameters. The loss function of the VAE is defined as following:

$$\mathcal{L}_{VAE} = \mathbb{E}_{q_\theta(z|x)}[\log q_\phi(x|z)] - \mathrm{KL}(q_\theta(z|x) \parallel p(z)), \tag{1}$$

where $\mathrm{KL}(. \parallel .)$ stands for the Kullback-Leibler divergence. By regularizing the posterior $q_\theta(z|x)$ with a prior over the latent representation $p(z) \sim \mathcal{N}(0, \mathbf{I})$, where $\mathbf{I}$ is identity matrix, VAE learns a latent representation $q_\theta(z|x)$ that contains the variations in the data. The goal of disentangled representation learning is to identify the latent representation $z \in \mathbb{R}^d$ where each latent variable only corresponds to one of the generative factors for given data $x$. To achieve this, $\beta$-VAE augments VAE objective with an adjustable hyperparameter $\beta$ as:

$$\mathcal{L}_{\beta-VAE} = \mathbb{E}_{q_\theta(z|x)}[\log q_\phi(x|z)] - \beta\mathrm{KL}(q_\theta(z|x) \parallel p(z)). \tag{2}$$

The addition of $\beta$ encourages the posterior $q_\theta(z|x)$ to match the factorized unit Gaussian prior $p(z)$. It enhances the independence among the latent variables thus disentangling the representation. On the other hand, it reduces the amount of information about $x$ stored in $z$, which can lead to a poor reconstruction especially for high values of $\beta$. This trade-off is further discussed from the rate-distortion theory perspective in Burgess et al. (2017).

To reduce the trade-off, different augmentations of $\beta$-VAE objective are proposed Burgess et al. (2017); Kim & Mnih (2018); Chen et al. (2018); Kumar et al. (2017). Locatello et al. (2018) categorized these methods into three main categories of *bottleneck capacity*, *penalizing the total correlation* and *disentangled priors*. Burgess et al. (2017) focused on *bottleneck capacity* and proposed to gradually increase the average KL divergence from zero for each generative factor. This method relaxes the information bottleneck during training via increasing the encoding capacity through a parameter $C$ that is linearly dependent on the training iteration. Kim & Mnih (2018) aimed

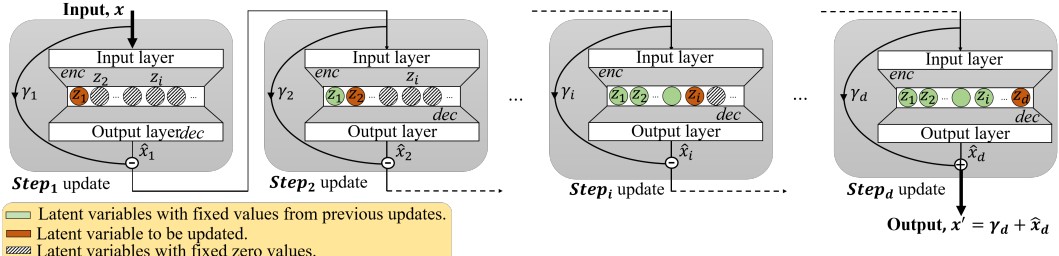

**Figure 1:** "Residual learning" mechanism consists of $d$ steps in a single forward pass with the same encoder $q_\theta(z|x)$ and decoder $q_\phi(x|z)$. Latent variables are sequentially sampled from the encoder. In step $i$, only the $i$th latent variable $z_i$ follows the distribution learned from the current residual. Previous latent variables follow the same distribution learned from their corresponding residuals. The latent variables $z_{i+1}$ to $z_d$ have fixed 0 value. The final output $x'$ consists of the decoder output using all the latent variables $\hat{x}_d$ and the skip connection $\gamma_d$.

to solve the disentangled representation learning by *minimizing the total correlation (TC)* term. They proposed FactorVAE where the objective is augmented with a TC term controlled by hyperparameter $\gamma$. Minimizing the TC term forces the hidden representation to be factorial and hence independent. Chen et al. (2018) looked into an alternative way to minimize the TC term in the augmented objective, named $\beta$-TCVAE. They used a mini-batch based alternative instead of a density-ratio-trick based method from FactorVAE. However, the results from both methods are sensitive to hyperparameter associated with the TC term. Kumar et al. (2017) studied the *disentangled prior* and introduced a regularizer to the objective that is associated with hyperparamter $\lambda$. This regularizer encourage the covariance of $q_\phi(z)$ to match the identity matrix. While all aforementioned approaches have shown promising results, they rely on a careful tuning of the hyperparamater introduced in the augmented objective functions such as $\beta$ in Higgins et al. (2017), $C$ in Burgess et al. (2017), $\gamma$ in Kim & Mnih (2018) and Chen et al. (2018), and $\lambda$ in Kumar et al. (2017). Finding the optimal hyperparamater setting can be challenging especially in an unsupervised learning setting where the evaluation of the results mainly relies on visualization and human inspection. More importantly, Locatello et al. (2018) found that hyperparameter tuning is more important for state-of-the-art performance than the choice of augmented objective functions.

In this work, we propose SR-VAE to address the aforementioned challenge. Instead of the forward pass in the original VAE, SR-VAE uses a "Residual learning" forward pass illustrated in Fig. 1 as an inductive bias on the model. SR-VAE consists of two components: 1) *Explicit dependency in the latent space* via a multi-step sequential forward pass where one latent variable is updated at each step; 2) *Decomposition of the reconstruction* via skip connetion between the input and network output to relax the network reconstruction constraint. Together, these two components enable SR-VAE to address the trade-off between the reconstruction and disentanglement in VAE using the original objective in Eq. 1. In the next section, we first describe the details of the "Residual learning" forward pass and SR-VAE training. We then discuss the two aforementioned components in detail. In Section 5, we demonstrate the effectiveness of SR-VAE and investigate the effect of each component separately. The experimental results show our approach achieves better disentanglement and reconstruction results compared to the current state-of-the-art approaches, including $\beta$-VAE Higgins et al. (2017) and FactorVAE Kim & Mnih (2018).

# 3 SEQUENTIAL RESIDUAL VARIATIONAL AUTOENCODER – SR-VAE

Same as the original VAE, SR-VAE consists of an encoder network noted as $q_\theta(\vec{z}|x)$, and a decoder network noted as $q_\phi(x|\vec{z})$. Here $x$ and $\vec{z}$ stand for input data and latent representation vector; $\theta$ and $\phi$ represent encoder and decoder network parameters. Let the dimension of the latent representation be $d$, SR-VAE learns $\vec{z} = [z_1, z_2, \ldots, z_d] \in \mathbb{R}^d$ as the latent representation of the data. Its forward pass follows a "Residual learning" mechanism that consists of $d$ steps. Each step only updates one latent variable. In the first step, the input data $x$ passes through the encoder to calculate the parameterized posterior, noted as $\vec{\mu}_1$ and $\vec{\sigma}_1$. Instead of drawing samples for all latent variables $\vec{z} \sim \mathcal{N}(\vec{\mu}_1, \vec{\sigma}_1)$, we only sample the first latent variable $z_1 \sim \mathcal{N}(\vec{\mu}_1[1], \vec{\sigma}_1[1])$ and set the remaining latent variables to 0. The modified latent variables $\vec{z} = [z_1, 0, \ldots, 0]$ then passes the decoder to generate the output noted

**Algorithm 1** SR-VAE Forward Pass

**Input:** observation $x$, latent dimension $d$, VAE encoder $q_\theta(z|x)$, VAE decoder, $q_\phi(x|z)$

**Output:** reconstruction $x'$, latent representation $\vec{\mu}', \vec{\sigma}'$

1: $\gamma_1 \leftarrow x$
2: $\vec{\mu}' = [0, \ldots, 0] \in \mathbb{R}^d$
3: $\vec{\sigma}' = [0, \ldots, 0] \in \mathbb{R}^d$
4: **for** $i = 1$ **to** $d$ **do**
5:      $\{\vec{\mu}_i, \vec{\sigma}_i\} \leftarrow$ *Encoder:* $q_\theta(\gamma_i)$
6:      $\vec{\mu}'[i] = \vec{\mu}_i[i]$
7:      $\vec{\sigma}'[i] = \vec{\sigma}_i[i]$
8:      $\vec{z} \leftarrow$ *Reparameterize($\vec{\mu}', \vec{\sigma}'$)*
9:      $\hat{x}_i \leftarrow$ *Decoder:* $q_\phi(\vec{z})$
10:      **if** $i < d$ **then**
11:          $\gamma_{i+1} \leftarrow \gamma_i - \hat{x}_i$
12:      **end if**
13: **end for**
14: $x' \leftarrow \hat{x}_d + \gamma_d$

**Algorithm 2** SR-VAE Learning

**Input:** Dataset $X$, batch size $m$, latent dimension $d$, Initialize VAE parameters $\theta, \phi$

1: **repeat**
2:      *Randomly select batch* $\mathbf{x} = \left(x^{(i)}\right)_{i \in \mathbf{B}}$ *of size* $m$,
3:      $\{\mathbf{x}', (\vec{\mu}, \vec{\sigma})\} \leftarrow$ Forward_pass($\mathbf{x}$)
4:      $\mathcal{L}_{recon} \leftarrow MSE\_loss(\mathbf{x}, \mathbf{x}')$
5:      $\mathcal{L}_{\mathcal{KL}} \leftarrow -\frac{1}{2} \sum_{j=1}^{\mathbf{B}} \sum_{i=1}^{d}$
         $[1 + \log(\sigma^j[i])^2 - (\mu^j[i])^2 - (\sigma^j[i])^2]$
6:      $\mathcal{L} \leftarrow \mathcal{L}_{\mathcal{KL}} + \mathcal{L}_{recon}$
7:      $\{\theta, \phi\} \leftarrow Backward(\mathcal{L})$
8: **until** convergence of objective

as $\hat{x}_1$. We subtract the decoder output from the skip connection (defined as an Identity function) as the input for the second pass, noted as $\gamma_2 = \gamma_1 - \hat{x}_1$. In the second pass, $\gamma_2$ passes the *same* encoder to generates a new parameterized posterior ($\vec{\mu}_2$ and $\vec{\sigma}_2$). This time, we sample only the second latent variable with this parameterized posterior as $z_2 \sim \mathcal{N}(\vec{\mu}_2[2], \vec{\sigma}_2[2])$. We re-sample the first latent variable with $z_1 \sim \mathcal{N}(\vec{\mu}_1[1], \vec{\sigma}_1[1])$ while setting the remaining latent variables to 0. The modified latent variable $\vec{z} = [z_1, z_2, 0, \ldots, 0]$ is then used to generate the new reconstruction $\hat{x}_2$. We then calculate the corresponding residual $\gamma_3 = \gamma_2 - \hat{x}_2$ as the input for the third pass. In the $i$th pass, the $i$th latent variable is sampled from the encoder thus $z_i \sim \mathcal{N}(\vec{\mu}_i[i], \vec{\sigma}_i[i])$. The previous updated latent variables follow their corresponding residual encoding and the remaining latent variables are set to zeros, $\vec{z} = [z_1, z_2, \ldots, z_i, 0, \ldots, 0]$. The process repeats $d$ times such that all the latent variables are sampled. In step $d$, the final output of SR-VAE, $x'$ consists of the decoder output $\hat{x}_d$ and the residual term $\gamma_d$ as $x' = \hat{x}_d + \gamma_d$. In the case where $d = 1$, SR-VAE follows the last step and the input is connected to the output through the skip connection. Algorithm 1 shows the pseudo code of the "Residual learning" forward pass in SR-VAE.

We train the SR-VAE with the original VAE objective defined in Eq. 1. The parameters are updated using the standard back-propagation demonstrated in Algorithm 2. The prior $p(z)$ is set to the isotropic unit Gaussian $\mathcal{N}(0, \mathbf{I})$ and posterior $q_\theta(z|x)$ is parameterized as Gaussians with a diagonal covariance matrix. The "reparametrization trick" is used to transform each random variable $z_i \sim q_\theta(z|x)$ as a differentiable transformation of a noise variable $\epsilon \sim \mathcal{N}(0, 1)$ with $z_i = \mu_i + \sigma_i \epsilon$.

Due to the sequential update process, SR-VAE can generate a sequence of images during the forward pass. As we shall see in Section 5, these images reflect image transformations corresponding to the disentangled factors at different steps. Comparing to other VAE based approach that directly generates a single reconstruction output by sampling from the joint distribution $\vec{z} \sim q_\theta(\vec{z}|x)$ for all latent variables, this step-by-step visual inspection allows for better understanding of the learned generative factor. As a result, SR-VAE provides a new way to understand the disentangled representation results.

**Explicit Dependency in the Latent Space:** The SR-VAE forward pass defines a sequential update of latent variables: the added latent variable $z_i$ at step $i$ learns from the residual between the input data and the previously updated latent variables $z_j, \forall j \in \{1, \ldots, i-1\}$. This procedure defines explicit dependency among the latent variables in the posterior that can be written as $q_\theta(z_1, z_2, ..., z_d|x) = q_\theta(z_1|x)q_\theta(z_2|z_1, x)...q_\theta(z_d|z_1, ..., z_{d-1}, x)$. The KL loss term of the original VAE objective in Eq. 1 encourages the posterior $q_\theta(z_1, z_2, ..., z_d|x)$ to match the factorized unit Gaussian prior $p(\vec{z})$. Adding the explicit dependency by the "Residual learning" mechanism, the SR-VAE objective can be seen as a modified VAE objective:

$$\underset{\theta, \phi}{\text{maximize}} \quad \mathcal{L}_{SR-VAE} = \mathbb{E}_{q_\theta(\vec{z}|x)}[\log q_\phi(x|\vec{z})] - \text{KL}(q_\theta(\vec{z}|x) \parallel p(\vec{z})),$$

$$\text{subject to} \quad p(z_1) \sim q_\theta(z_1|x), p(z_2) \sim q_\theta(z_2|z_1, x), \ldots, p(z_d) \sim q_\theta(z_d|z_1, ..., z_{d-1}, x).$$

(3)

These constraints encourage the newly added latent variable to be independent of the ones already added, thus enhance the disentanglement of the latent representation. Moreover, the solution space of Eq. 3 is a subset of the original VAE. The constrained objective limits the optimization search space to regions where a better local optimal solution exists. We empirically verify this result in terms of both the performance and stability to random initialization in Section 5.

**Decomposition of the Reconstruction:** The final output of SR-VAE, $x'$ consists of the decoder output and the residual term as $x' = \hat{x}_d + \gamma_d$. This formulation relaxes the reconstruction constraint on the network's decoder output when comparing with other VAE-based approaches. More importantly, it creates a balancing measure between the data generation and reconstruction. In one extreme case, the input $x$ directly passes through the first $d-1$ steps and reaches step $d$ as the input. In this case, SR-VAE becomes the original VAE model with added skip connection between input and output (see the last step in Fig 1). This architecture relaxes the VAE reconstruction hence reduces VAE reconstruction and disentanglement trade-off. We will show in Section 5.1, this architecture alone can reach similar performance to FactorVAE. On the other extreme case, if the first $d-1$ steps have learned a perfect disentangled representation of the data, the input for step $d$ would be 0. In this case, the reconstruction loss encourages SR-VAE to generate the input data from learned latent representation vectors. Combining these two extreme cases, SR-VAE can be understood as a training mechanism to balance between a VAE model with emphasis on the reconstruction quality (as the first case) and the data generation model given the learned latent variables (as the latter case).

Notice that each of the aforementioned components can be separately added to VAE as a modified model. To add the *explicit dependency in the latent space* component, we can apply the sequential forward pass of SR-VAE with the output $x' = \hat{x}_d$. We refer to this model as SeqVAE. For the *Decomposition of the Reconstruction* component, as mentioned earlier, it is equivalent to adding a skip connection to the original VAE between the input and output. We refer to this model as ResVAE. Using these two models, we perform the ablation study to understand the effectiveness of each individual component in Section 5.1.

**Computational Complexity:** SR-VAE replaces the standard forward pass of VAE with $d$ forward passes, thus increases the computational complexity. However, in addition to the improved state-of-the-art performance, it eliminates the hyperparameter tuning associated with prior works. As mentioned earlier, the hyperparameter tuning was shown to be critical for state-of-the-art performance. It is a difficult and time-consuming process especially for unlabeled data due to: 1) The large hyperparameter search space of continuous values; 2) The lack of evaluation metric. As a result, we believe that the increased computational complexity by SR-VAE is reasonable. Moreover, we will show that each of the $d$ forward passes in SR-VAE correspond to a disentangled generative factor. Visualization of these intermediate steps provides a new way to understand the result.

## 4 RELATED WORK

**Connection to Other VAE-based Approaches:** We highlight the similarity and advantages of SR-VAE over the VAE-based approaches introduced in Section 2. The sequential update of latent variables in SR-VAE is similar to the idea of gradually increasing the KL divergence in Burgess et al. (2017). Instead of introducing an augmented objective, SR-VAE directly achieves this by learning one latent variable at a time. When comparing to the work in Kumar et al. (2017), SR-VAE encourages the independence among the latent variables by defining an explicit latent variable dependency rather than emphasizing on individual statistics (the covariance between the latent representations in Kumar et al. (2017)). Finally, the explicit latent variable dependency defined by SR-VAE also encourages the factorial latent representation, serving the same purpose as lowering the TC term in Kim & Mnih (2018); Chen et al. (2018). It is worth noticing that the low TC term is necessary but not sufficient for disentangled representation.

**Connection to Residual Deep Neural Network:** ResNet He et al. (2016) introduces the idea of learning from residuals by adding the skip connections between layers such that input can propagate through layers. The key idea of ResNets is to replace learning the direct mapping between input and output, $H(x) = x \to y$, with learning a residual formulation, $H(x) = F(x) + x \to y$, where $F(x)$ represents stacked non-linear layers. This formulation reduces the loss of important information while propagating through the network. In addition, it was suggested that learning the residual mapping is easier compared to learning the direct mapping He et al. (2016). The proposed SR-VAE shares similar

skip connection structure as ResNets. Here $F(x)$ represents $VAE_i$ in SR-VAE. As in ResNets, $F(x)$ can learn useful abstraction of data while the skip connection $\gamma_i$ allows for circumventing difficulties in reconstructing the input data.

**Connection to Deep Recurrent Attentive Writer (DRAW):** DRAW Gregor et al. (2015) uses a sequential variational auto-encoding framework to achieve iterative construction of an image. DRAW deploys the recurrent neural network with the attention mechanism to dynamically determine where and what to generate. The attention mechanism serves a similar purpose to the skip connection in the "Residual learning" mechanism. Moreover, the idea of successively adding the decoder output for image generation in DRAW is similar to the reconstruction decomposition in SR-VAE. One main difference between the two approaches is that DRAW relies on the recurrent network framework to model the iterative image generation in the image space. SR-VAE uses the latent dependency to emphasize iterative generation of image in the latent space.

## 5 EXPERIMENTS

We compare SR-VAE with $\beta$-VAE and FactorVAE on four different datasets both quantitatively and qualitatively. The datasets used in this study includes: **2D Shape** Higgins et al. (2017), **Teapots** Eastwood & Williams (2018), **CelebA** Liu et al. (2014) and **Chairs** Aubry et al. (2014). Appendix A introduces the details of these datasets. For all datasets, we use *visualization* for qualitative evaluation by observing the changes in the reconstruction while altering only one latent dimension, known as the traversal of the latent variable. A good disentangled representation reveals interpretable changes in the reconstruction image corresponding to one generative factor. Moreover, **2D Shape** and **Teapots** datasets contain ground truth generative factors that are used to synthesize the dataset, which allow us to conduct quantitative evaluations. To compare with the previous studies, we use the metric proposed in Kim & Mnih (2018) (noted as *FactorVAE metric*) for **2D Shape**, and the metric proposed in Eastwood & Williams (2018) (noted as *Disentanglement-Informativeness-Completeness metric*)for **Teapots**. These two metrics are found to cover similar notions to other disentanglement metrics in Locatello et al. (2018). We implemented our approach using Pytorch Paszke et al. (2017), with the experiments run on several machines each with 4 GTX1080 Ti GPUs. See Appendix C for details on model architecture.

### 5.1 QUANTITATIVE EVALUATION

**Metrics:** The *FactorVAE metric* in Kim & Mnih (2018) is calculated as follows: 1) select a latent factor $k$, 2) generate new data $y$ with factor $k$ fixed and other factors varying randomly, 3) calculate the mean of $q_\theta(z|x)$, 4) normalize each dimension by its empirical standard deviation over all the data or a large enough subset, 5) build a majority-vote classifier with the input of index of the dimension with the lowest variance and the output of factor $k$. The classifier accuracy is used as the evaluation metric. Eastwood & Williams (2018) defines three criteria of disentangled representation, namely *disentanglement*, *completeness* and *informativeness*. Disentanglement is the degree to which the learned representation disentangles the underlying generative factors; completeness is the degree to which the generative factors are captured by one latent representation; and finally the informativeness is the amount of information of the generative factors that is captured by the latent representation. Disentanglement and completeness can be perceived by visualizing rows and columns of the Hinton diagram; and informativeness is calculated based on the mapping error between the learned latent representation and the ground truth factors.

**Comparison to $\beta$-VAE and FactorVAE:** Similar to previous studies, we set $d = 10$ for all datasets except for CelebA where $d = 32$ due to its complexity. We use the optimal parameter setting from the original studies in Higgins et al. (2017); Eastwood & Williams (2018) for $\beta$-VAE and FactorVAE. Fig. 2(a) and 2(b) show that SR-VAE outperforms $\beta$-VAE and FactorVAE in terms of both the reconstruction error and the disentanglement metric in Kim & Mnih (2018) on **2D Shape**. The best mean disentanglement measurement of SR-VAE is around 0.86, significantly higher than the one for $\beta$-VAE at 0.72 and FactorVAE at 0.81. For reconstruction error, $\beta$-VAE and FactorVAE converge to similar results while SR-VAE achieves better performance. In Fig. 2(c)-(e), we compare SR-VAE with $\beta$-VAE and FactorVAE using metric proposed in Eastwood & Williams (2018) on **Teapots**. Three criteria used in this metric are disentanglement, completeness and informativeness. Note that in the informativeness term in this metric is highly dependent to the regressor type. In Eastwood &

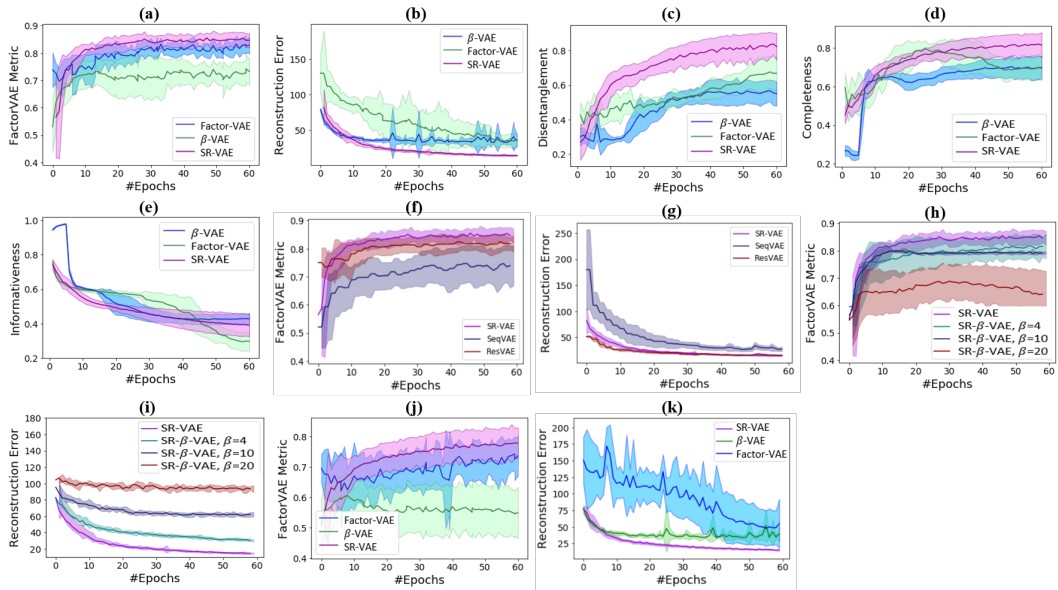

**Figure 2:** Quantitative evaluations. Similar to previous studies, all results are reported on the best 10 of 30 runs with random seeds except for (j) and (k). The line and shaded area correspond to the mean and confidence intervals. (a) and (b): the metric in Kim & Mnih (2018) and the reconstruction error for **2D Shape**; (c), (d) and (e): the three metrics in Eastwood & Williams (2018) for **Teapots**; (f) and (g): ablation study comparing the SeqVAE, ResVAE and SR-VAE using the *FactorVAE metric* and reconstruction error on **2D Shape**; (h) and (i): comparison between SR-VAE and SR-$\beta$-VAE, using the *FactorVAE metric* and reconstruction error on **2D Shape**; (j) and (k): Sensitivity to initialization study that compares the worst 10 of 30 runs with random seeds using the *FactorVAE metric* and reconstruction error on **2D Shape**.

Williams (2018) Lasso and Random Forest regressors are used which resulted in a different ordering of methods in informativeness score. Random Forest is used in our experiments to be comparable with the original paper. The results show SR-VAE achieves higher disentanglement and completeness scores compare to $\beta$-VAE and FactorVAE, while FactorVAE achieves the best informativeness. However, we believe the informativeness can be different if a more complex regressor was used.

**Ablation Study:** To investigate the individual effect of the two main components in SR-VAE as discussed in Section 3, we compare the performance among SeqVAE, ResVAE and SR-VAE on **2D Shape** dataset. Similar as before, the top 10 of 30 runs with random initialization of all models are reported in Fig. 2(f) and 2(g). The results show that both ResVAE and SeqVAE perform worse than SR-VAE in terms of *FactorVAE metric*. When comparing with $\beta$-VAE and FactorVAE, ResVAE performs similar to FactorVAE while SeqVAE performs similar to $\beta$-VAE. One interesting result we noticed is that the reconstruction error from ResVAE is similar to if not better than SR-VAE. These results verify our analysis in Section 3 that the *decomposition of reconstruction* relaxes the reconstruction constraint of the network, and adding the *explicit dependency in the latent space* improves disentangled representation learning. While both components are important for the superior performance of SR-VAE, relaxing the construction constraint on the network with skip connection is more important as it directly addresses the bottleneck of VAE.

**SR-VAE with $\beta$-VAE objective:** We also examined if using the $\beta$-VAE objective in Eq. 2 with the "Residual learning" mechanism would improve the performance, referred to as SR-$\beta$-VAE. If so, the proposed "Residual learning" mechanism would benefit from the augmented objective to achieve better performance. Figures 2(h) and 2(i) show that best disentanglement score is obtained by SR-VAE and higher $\beta$ values do not help with improving the performance. These results further verify the effectiveness of SR-VAE in solving the trade-off between disentanglement and reconstruction in VAE-based approaches.

**Sensitivity to initialization:** The study in Locatello et al. (2018) showed that existing approaches are sensitive to the initialization in addition to the hyperparameter tuning. One advantage of SR-VAE is that it reduces the solution space and improve training stability. To verify this, we compare the worst

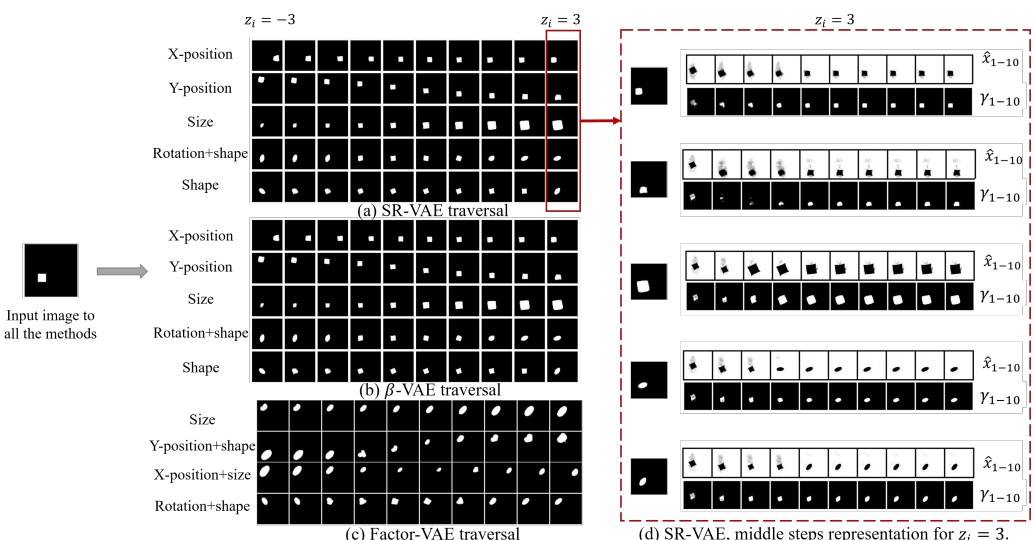

**Figure 3:** (a)-(c) Latent traversals with the same input image across each latent dimension with $d = 10$ for **2D Shape** dataset, using SR-VAE, $\beta$-VAE and FactorVAE respectively ; (d) Decomposition of decoder output and skip connection at each step in SR-VAE during latent traversal towards the last column of (a).

10 runs out of 30 for **2D Shape** in Fig 2(j) and Fig 2(k). We consistently observe better performance and smaller variances by SR-VAE, suggesting its robustness against random initialization.

## 5.2 QUALITATIVE EVALUATION

Figure 3(a)-(c) show the latent traversal of SR-VAE, $\beta$-VAE and FactorVAE for a fixed input image of **2D Shape**. $z_i$ values are chosen from the range of -3 to 3 as shown in the figure. We see that while all three models are capable of finding data generative factors – *X-position, Y-position, Shape, Rotation and Scale* – $\beta$-VAE and FactorVAE struggle to disentangle these factors completely. As mentioned in Kim & Mnih (2018) shape is a discrete variation factor in **2D Shape**. Ideally, this factor should be modeled with a discrete rather than Gaussian latent variable. Despite this mismatched assumption, SR-VAE still captures the shape in the fifth latent variable. However, it also mixes the size with the shape between oval and square in the third latent variable. We also experiment the latent traversal with the **Teapots** dataset and observe superior performance as shown in Appendix D.

For datasets without ground truth generative factors, such as **CelebA** and **Chairs**, inspecting latent traversals is the only evaluation method. Similar as before, we used the optimal parameter setting for $\beta$-VAE and FactorVAE from the original studies in Higgins et al. (2017); Eastwood & Williams (2018). As seen in Figure 4 for CelebA dataset, SR-VAE is able to learn interpretable factors of variation such as background, face and hair characteristics, skin color, etc. Compared to $\beta$-VAE and FactorVAE, we observe some common factors as well as some unique ones. Note that only the most obvious factors are presented in this figure. Moreover, the interpretation of each latent dimension is based on our best judgment. We also observe better reconstruction quality with more details using SR-VAE method. Reconstruction losses also confirm this observation with the converged values of 300, 252 and 158 for $\beta$-VAE, FactorVAE and SR-VAE, respectively. Admittedly, careful tuning of parameters in $\beta$-VAE and FactorVAE could potentially reveal more latent variables. However, finding the optimal value is a difficult task especially when there is no prior information about the generative factors of data and a quantitative metric is not available.

**Visualizing the "Residual learning" Mechanism:** To gain a better understanding of the internal process of the "Residual learning" mechanism, we show the decoder output, the residual mapping, of each internal step ($\hat{x}_1, ..., \hat{x}_{10}$) and their skip connections ($\gamma_1, ..., \gamma_{10}$) for **2D Shape** in Fig. 3(d). Each row presents the internal steps when setting different latent variables ($z_1$ to $z_5$) to value 3. The final outputs of this process correspond to the last column of Fig. 3(a). In this figure, we observe the step by step transition to the final transformed image during the latent traversal. The result shows that

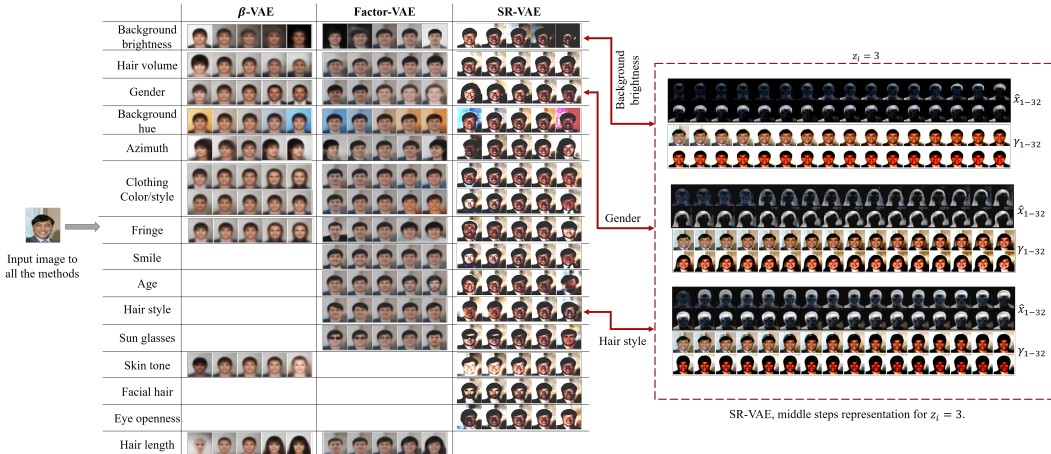

**Figure 4:** *Left:* Latent traversals across each latent dimension with $d = 64$ for **CelebA** using SR-VAE, FactorVAE and $\beta$-VAE, respectively. *Right:* Decomposition of decoder output and skip connection at each step in SR-VAE during latent traversal towards the first column of the corresponding row.

the two terms are working together to capture the learned disentangled factor at each step. Based on Fig. 3(a), we know the learned factors in each step are: *X-position, Y-position, Size, Rotation+shape*, and *Shape*, respectively. In Fig. 3(d), we observe that X-position of the reconstructed image are generated during the first step. In step two, both X-position and Y-position are generated. This process continues and at each step the decoder output and the residual transform the image according to the learned latent encoding.

Similarly, we show the step-by-step visualization for **CelebA** dataset along with its latent traversal result in Fig 4. We highlight a few factors due to the space limit. Although **CelebA** presents challenge in the complexity of real-world image, we observe similar results as the **2D Shape** dataset. The step-by-step visualization shows how the latent factors are related to the transformed face image during the latent traversal. For example, the gender factor can be identified as the fifth latent factor as we observe major changes in the eyes and hair style from step five. Another example is the background contrast factor where major changes can be observed in step eight. These step-by-step visualizations provide an alternative way to understand and interpret the learned disentangled factors and can be interesting for data generation tasks.

## 6 Conclusions

In this work, we propose SR-VAE for disentangled representation learning in an unsupervised setting. The proposed solution defines the "Residual learning" mechanism in the training regime, instead of augmented objective, to solve the trade-off between disentanglement and reconstruction of the VAE-based approaches. SR-VAE defines explicit dependency structure between latent variables and decomposes the reconstruction via skip connection. We showed that SR-VAE achieves state-of-the-art results compared to previous approaches including $\beta$-VAE and FactorVAE. Moreover, SR-VAE can be directly applied to any VAE architecture without an additional hyperparameter tuning. The step-by-step process of the SR-VAE provides novel ways to visualize the results and understand the internal process of learning disentangled factors. We believe this can open a new direction for future research towards disentangled representation learning.

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

## A    DATASETS

**2D Shape** Higgins et al. (2017) is a synthetically generated dataset with ground truth generative factors. It contains $737, 280$ binary 2D Shapes of heart, oval and square of size $64 \times 64$ images. There are five independent data generative factors includeing shape, position X, position Y, scale and rotation.

**Teapots** Eastwood & Williams (2018) is another synthetically rendered data with ground truth generative factors. It consists of $200, 000$ images of a teapot object of size $64 \times 64$ with different colors and poses. The camera is centered on the object and the background is removed. Five different generative factors are independently sampled from their corresponding uniform distribution: azimuth $\sim U[0, 2\pi]$, elevation $\sim U[0, \pi/2]$, red $\sim U[0, 1]$, green $\sim U[0, 1]$ and blue $\sim U[0, 1]$.

**CelebA** Liu et al. (2014) is a real-image dataset without ground truth generative factors. It contains ten thousand different celebrity identities, each with twenty images. Each image is annotated with forty face attributes such as: young, pale skin, bangs, straight hair, no beards, etc. The cropped $64 \times 64 \times 3$ face images are used in this study as used in several earlier works.

**Chairs** Aubry et al. (2014) includes $86, 366$ 3D chair images rendered of 3D CAD models downloaded from Google/Trimble 3D Warehouse. This dataset also comes without ground truth generative factors.

## B    MODEL ARCHITECTURE

Table 1 shows the encoder and the decoder architecture used in the experimental section, which is the same as in original $\beta$-VAE. For a fair comparison, similar network architecture is used in all the methods and no parameter tuning is done as for number of layers, nodes, non-linearity and etc.

## C    TRAINING DETAILS

Table 2 shows the training details which are used throughout the experiments. No tuning is done on these parameters. Table 3 also presents the hyperparameters used in the experiments for **Teapots** data and metric Eastwood & Williams (2018). No tuning is done on the main network and regressor parameters. Regressor parameters are similar to the ones used in Eastwood & Williams (2018). Parameters that are not listed in this table are similar to Table 2.

## D    MORE LATENT TRAVERSALS RESULTS

Figures 5 and 6 show the latent traversals of **Chairs** and **Teapots** datasets. As seen in Figure 5, SR-VAE can find more disentangled factors compared to $\beta$-VAE and Fcator-VAE methods. Three azimuth factors in Figure 5a cover different angles in the space and so they are disentangled. Figure 6 also shows the superior performance of our method in disentangling factors for **Teapots** dataset. Interestingly, both SR-VAE and Factor-VAE methods disentangle color factor in 3 latent representations. Each of these latent representations cover different range of color spectrum and hence they are disentangled.

**Table 1:** Encoder and Decoder architecture, *z-dim*: dimension of the latent representation; *nc*: number of input image channel.

| Encoder | Decoder |
| --- | --- |
| Input 64×64 binary/RGB image | Input $\mathbb{R}^{z-dim}$ |
| 4×4 conv, 32 ReLu, stride 2, pad 1 | FC $z - dim$×256, ReLu |
| 4×4 conv, 32 ReLu, stride 2, pad 1 | 4×4 upconv, 64 ReLu, stride 1 |
| 4×4 conv, 64 ReLu, stride 2, pad 1 | 4×4 conv, 64 ReLu, stride 2, pad 1 |
| 4×4 conv, 64 ReLu, stride 2, pad 1 | 4×4 conv, 32 ReLu, stride 2, pad 1 |
| 4×4 conv, 256 ReLu, stride 1 | 4×4 conv, 32 ReLu, stride 2, pad 1 |
| FC 256 × (2×$z - dim$) | 4×4 conv, nc , stride 2, pad 1 |

**Table 2:** Hyperparameters setting.

| Parameter | value |
| --- | --- |
| Batch size | 64 |
| Latent dimension | 10 |
| Optimizer | Adam |
| Adam: beta1 | 0.9 |
| Adam: beta2 | 0.999 |
| Learning rate | 1e-4 |
| Decoder type | Bernoulli |

**Table 3:** Hyperparameters setting for metric Eastwood & Williams (2018) experiments on **Teapots**.

| Parameter | value |
|---|---|
| Lasso: $\alpha$ | 0.02 |
| Random Forest: #estimators | 10 |
| Random Forest: max-depth for 5 ground truth factors | [12, 10, 3, 3, 3] |
| Learning rate | 1e-5 |

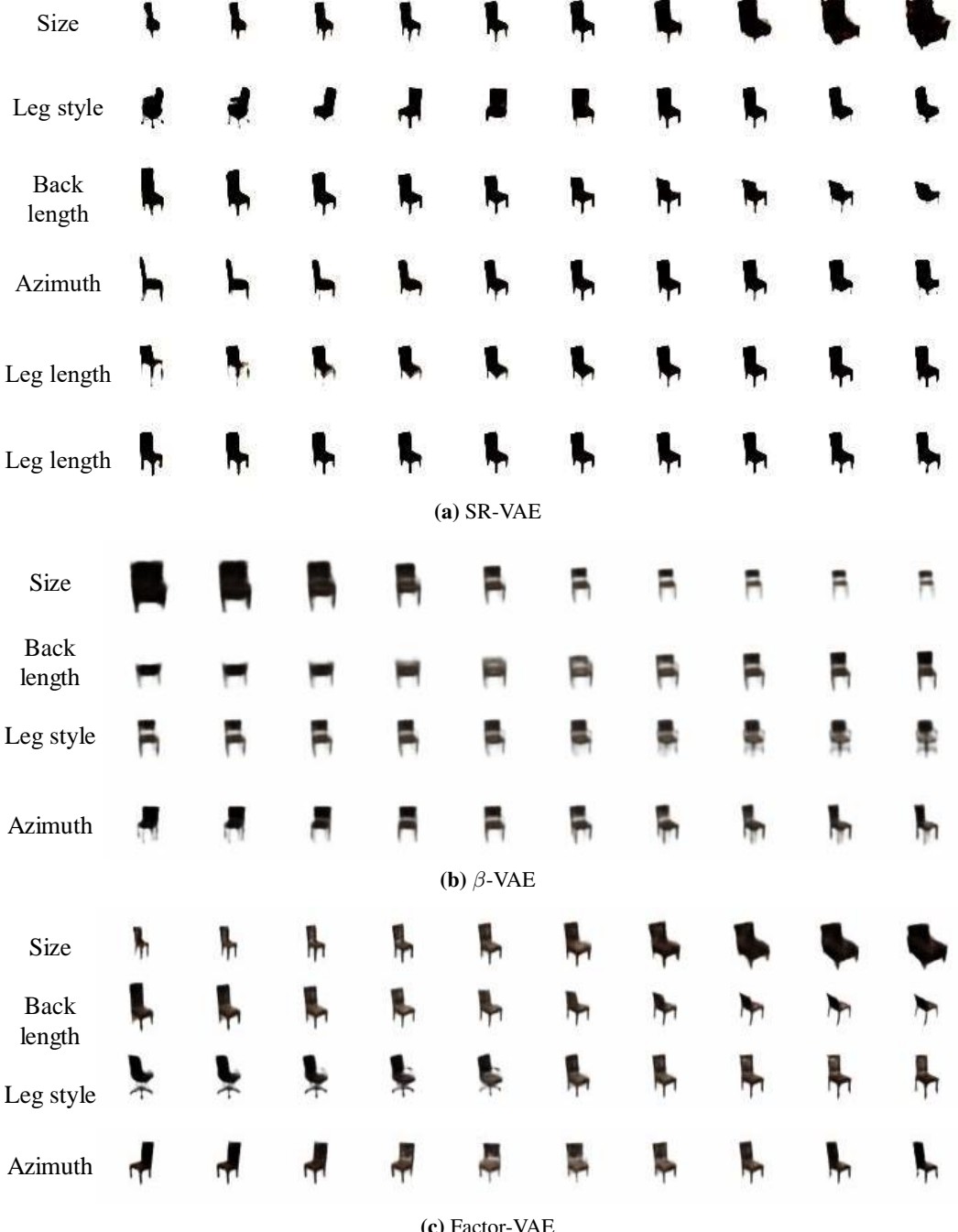

**Figure 5:** Latent traversals across each latent dimension where $d$ is set to 10 for **Chairs** dataset.

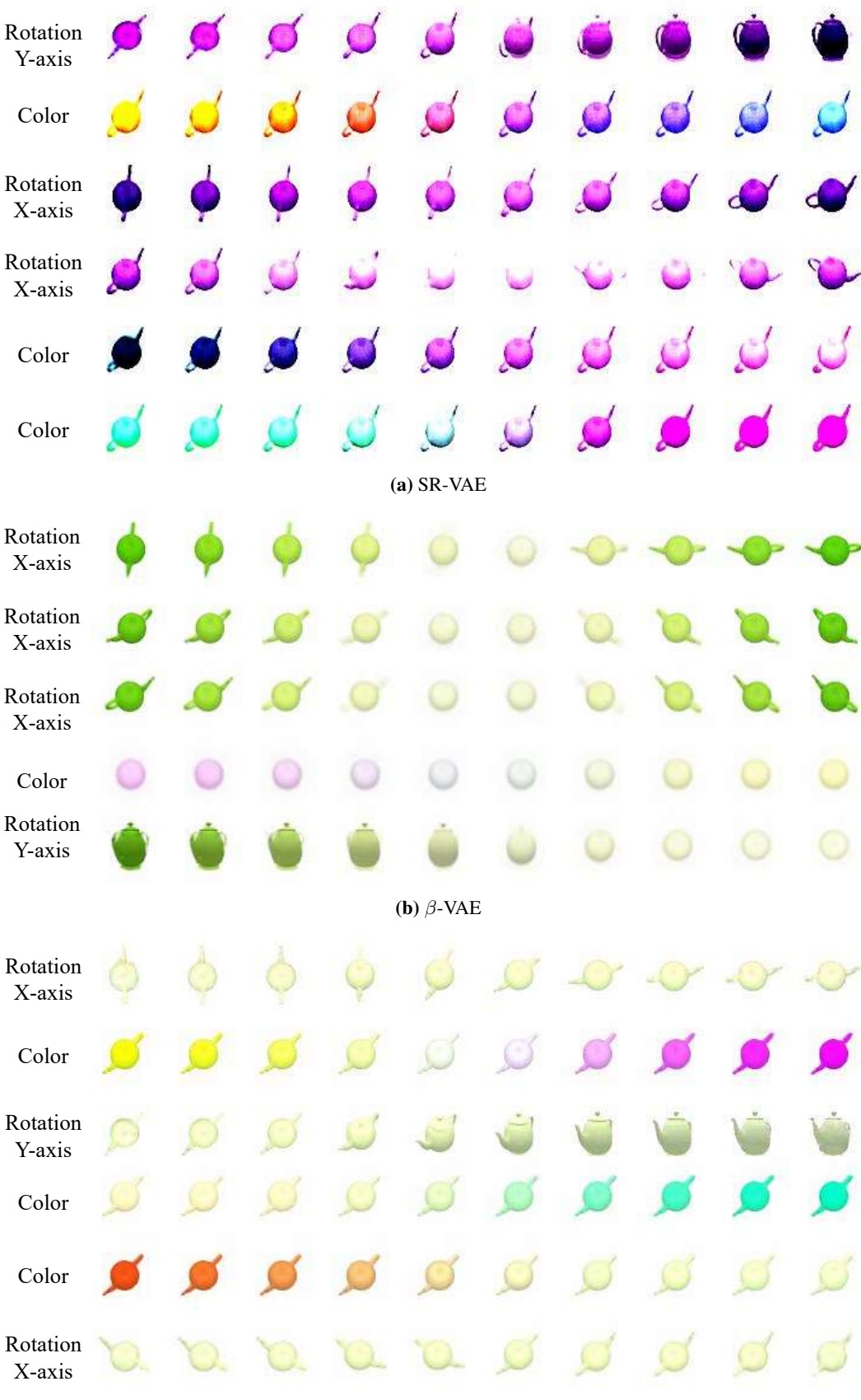

(a) SR-VAE

(b) $\beta$-VAE

(c) Factor-VAE

**Figure 6:** Latent traversals across each latent dimension where $d$ is set to 10 for **Teapots** dataset.

