# OpenReview forum: "Disentangled Representation Learning with Sequential Residual Variational Autoencoder"
_ICLR.cc/2020/Conference — Reject_

### Official Review · AnonReviewer2 · 2019-10-22
**Official Blind Review #2**

**Rating:** 8

**Review:**

The authors of this paper present a novel that for unsupervised disentangled representation learning. The model, named sequential residual VAE (SR-VAE), gradually activates individual latent variables to reconstruct residuals. Quantitative and qualitative experiments show that the proposed model outperforms beta-VAE and Factor-VAE.  Since the training involves a sequence of model training,  SR-VAE certainly consumes more time than other VAEs. Minors: citations in the main text should be put in brackets.

**Experience Assessment:**

I have published one or two papers in this area.

**Review Assessment: Checking Correctness Of Derivations And Theory:**

I carefully checked the derivations and theory.

**Review Assessment: Checking Correctness Of Experiments:**

I carefully checked the experiments.

**Review Assessment: Thoroughness In Paper Reading:**

I read the paper thoroughly.

---

> ### Author Response · Authors · 2019-11-06
> **Authors Response to Review #2**
>
> We understand the challenges of reviewing large volume of ICLR submissions and greatly appreciate the reviewer's time and effort to review our submission.
>
> We thank the reviewer for the positive feedback and acknowledging the novelty of the proposed approach to address an important yet challenging task of disentangled representation learning.
>
> The computation complexity of SR-VAE in a single training process is longer comparing to other approaches. However, as responded to another reviewer, the current state-of-art approaches relies heavily on hyperparameter tuning that requires hundreds runs, potential years of GPU times to figure out the optimal solution. The proposed approach eliminate this search process, thus reduce the total computation. More importantly, the lack of quantitative metric for unsupervised disentanglement learning makes it hard to understand the effect of hyperparameter tuning (the current common evaluation for unsupervised disentanglement learning is human visual inspection). We believe the proposed approach significantly reduce this effort by eliminating the hyperparameter search.
>
> As the reviewer suggested, we added bracket of the reference to increase the readability.

---

### Official Review · AnonReviewer3 · 2019-10-23
**Official Blind Review #3**

**Rating:** 3

**Review:**

Overview:
Authors introduce a new VAE-based method for learning disentangled representations.
The main idea is to apply a “residual learning mechanism”, which resembles an autoregressive model, but here conditioning between sequential steps is done in both latent and input spaces. Namely, for each input, the method involves making several (one per *each* embedding dimension) sequential passes, where each pass takes the residual between the input and the current accumulated output (i.e. pixel-wise difference between two images) as well as the values of the sampled embeddings so far.
Authors report quantitative and qualitative evaluation with a set of reasonable baselines (FactorVAE, beta-VAE), and the results seem to be slightly better.

Decision:
The writing quality is not great: there are frequent typos and not-so-precise formulations, and is at times hard to follow (see list below).
The method itself is not really well-motivated: there seems to be no formal justification provided, and the informal one is not very clearly explained in the paper.
Scalability of the method is clearly an issue. Many applications might require the size of embeddings to have hundreds of dimensions, which would mean that the given method cannot really be applied for non-toy problems.
With all those considerations in mind, I cannot recommend to accept this paper as is, thus the final rating, “reject”.

Additional comments / typos:
* It would be worth seeing how this approach relates to auto-regressive models.
* p2. “encoder maps … x to a latent representation q” - this sentence is not very strict.
* p2. “.. is defined as following”: as follows?
* p2. “.. in addition of b”
* p3. “.. this reguralizer encourage”
* p7: “… it reduces the solution space and improve training stability“: is there any sort of theoretical reasoning that could support this?
* The reference format is not very readable, probably better to change to (Names, Year).

<update>
Authors addressed some of my concerns (thanks!), thus I increased my rating slightly.
There is still a large concern wrt to the computational complexity. I kind of understand the argument about the hyperparameter tuning, but it seems not really fair to compare to a bruteforce parameter search (one can probably do some sort of bayesian optimization).
</update>




**Experience Assessment:**

I have published one or two papers in this area.

**Review Assessment: Checking Correctness Of Derivations And Theory:**

I assessed the sensibility of the derivations and theory.

**Review Assessment: Checking Correctness Of Experiments:**

I assessed the sensibility of the experiments.

**Review Assessment: Thoroughness In Paper Reading:**

I read the paper thoroughly.

---

> ### Author Response · Authors · 2019-11-06
> **Authors Response to Review #3**
>
> We understand the challenges of reviewing large volume of ICLR submissions and greatly appreciate the reviewer's time and effort to review our submission.
>
> Motivation ---> We aimed to motivate our study in Section 2 by discussing the issues of disentanglement learning with VAE. The formal justification of these issues was carefully discussed in theory among the cited prior works including: Section 2 of “Disentangling by Factorising” [Kim & Mnih (2018)] and Section 4 of “Understanding disentangling in β-VAE” [Burgess et al.(2017)]. In addition, we would like to clarify the main issue of learning of disentangled representation with VAE lies in the trade-off between the information bottleneck and the input reconstruction. Current approaches use augmented objective to address this problem. We propose to address this issue with a different training regime. We’ve revised the manuscript in section 2 to cite the formal justification more clearly.
>
> Computational Complexity ---> We agree with reviewer that the computation complexity of a single training process is longer comparing to other approaches. However, as discussed in Section 3 of our paper and highlighted in [Locatello et al, 2018], the current state-of-art approaches relies heavily on hyperparameter tuning that requires hundreds runs, potential years of GPU times to figure out the optimal solution. The proposed approach eliminates this search process, thus reduces the total computation. More importantly, the lack of quantitative metric for unsupervised disentanglement learning makes it hard to understand the effect of hyperparameter tuning (the current common evaluation is human visual inspection). We believe the proposed approach significantly reduces this effort by eliminating the hyperparameter search.
> Regarding the comments “Many applications might require the size of embeddings to have hundreds of dimensions”, we agree that different applications require different latent space dimension. However, the idea of disentangled representation is to search for a compact (low dimension) representation space for real-world objects. This resembles human’s capability to understand an object in a concise manner. For applications that involves complicated components, many recent work (e.g. Multi-Object Representation Learning with Iterative Variational Inference,  MONet: Unsupervised Scene Decomposition and Representation, GENESIS: Generative Scene Inference and Sampling with Object-Centric Latent Representations) focus on decomposing the scene into different components/objects/parts where each individual component can be modeled by a low-dimension disentangled representation.
>
> Additional comments / typos --->
> * It would be worth seeing how this approach relates to auto-regressive models.
> The proposed SR-VAE does resemble the non-linear autoregressive model for inference with the latent representation. It can be understood as the future z values are predicted based on the past z values along with the accumulated reconstruction. We have added this discussion to the related work section in the paper.
>
> * p2. “encoder maps … x to a latent representation q” - this sentence is not very strict.
> It would be great if the reviewer could provide details on the confusion and we are happy to address it accordingly.
>
> * p7: “… it reduces the solution space and improve training stability“: is there any sort of theoretical reasoning that could support this?
> In general limiting the search space does not necessarily improve the optimization results unless better local optimal solutions are in the constrained space. Since the optimal solution of the original VAE or beta-VAE satisfies the dependency constraints introduced in SR-VAE, we believe SR-VAE could help with reaching a better local optimal solution. Empirically, we observed that SR-VAE improves training stability by comparing the worst 10 runs out of 30 for 2D Shape in Fig 2(j)(k). We consistently observe better performance and smaller variances by SR-VAE.
>
> We thank the reviewer for pointing out the typos and grammar errors. We have corrected these mistakes and carefully proof-read the paper. As another reviewer suggested, we also added bracket of the reference to increase the readability.

---

### Official Review · AnonReviewer1 · 2019-10-27
**Official Blind Review #1**

**Rating:** 3

**Review:**

The authors applied residual learning machenism in VAE learning, which I have seen such methods in Deep Generative Image Models using a Laplacian Pyramid of Adversarial Networks (https://arxiv.org/abs/1506.05751), which basically applied the residual learning method in GAN. But the authors fail to discuss the relationship and difference with this paper.
Also, the best paper from ICML 2019 claimed that unsupervised learning method can not really disentangle the features. They claim \beta-VAE, factor VAE is not good. The authors shall all discuess this point. Otherwise, it is not convincing to the readers.



**Experience Assessment:**

I have published one or two papers in this area.

**Review Assessment: Checking Correctness Of Derivations And Theory:**

I assessed the sensibility of the derivations and theory.

**Review Assessment: Checking Correctness Of Experiments:**

I assessed the sensibility of the experiments.

**Review Assessment: Thoroughness In Paper Reading:**

I read the paper at least twice and used my best judgement in assessing the paper.

---

> ### Author Response · Authors · 2019-11-06
> **Authors Response to Review #1**
>
> We understand the challenges of reviewing large volume of ICLR submissions and greatly appreciate the reviewer's time and effort to review our submission.
>
> --- Thank you for pointing out the reference paper which improves the image generation with Laplacian pyramid in GAN. As a general learning method, residual learning has been applied to different domains ranging from discriminative models (ResNet), generative model (GAN), and reinforcement learning (relatively recent Residual RL). While the suggested reference by the reviewer carries the idea of iterative learning process, we believe the closest study to our work is the framework of “DRAW” [Gregor et al. (2015)]. Notice that "DRAW" is also cited in the recommended paper. We have discussed in details the connection between our work and "DRAW" in section 4.
>
> --- We are glad that the reviewer is familiar with the work by Locatello et. al. in ICML 2019.  This work is heavily cited in our manuscript with its earlier arxiv version ([Locatello et al, 2018] in our paper). One of the key insights in this paper is: “theoretically prove that (perhaps unsurprisingly) the unsupervised learning of disentangled representations is fundamentally impossible without inductive biases both on the considered learning approaches and the data sets.”. This is indeed the motivation of our work. The proposed “Residual learning” forward pass serves as an inductive bias on the model to facilitate disentangled learning in VAE. It combines two important components of: 1) explicit dependency and 2) decomposition of reconstruction, to address the information bottleneck of disentanglement learning with VAE framework. To reduce the confusion, we updated our citation for  the work of Locatello et al from the arxiv version to the ICML version. We also modified Section 2 to emphasize this point.

---

### Decision · Program_Chairs · 2019-12-19

**Decision:**

Reject

**Comment:**

This paper that defines a “Residual learning” mechanism as the training regime for variational autoencoder. The method gradually activates individual latent variables to reconstruct residuals.

There are two main concerns from the reviewers. First, residual learning is a common trick now, hence authors should provide insights on why residual learning works for VAE. The other problem is computational complexity. Currently, reviews argue that it seems not really fair to compare to a bruteforce parameter search. The authors’ rebuttal partially addresses these problems but meet the standard of the reviewers.

Based on the reviewers’ comments, I choose to reject the paper.